# Quantum Battery Based on Hybrid Field Charging

**DOI:** 10.3390/e24121821

**Published:** 2022-12-14

**Authors:** Yunxiu Jiang, Tianhao Chen, Chu Xiao, Kaiyan Pan, Guangri Jin, Youbin Yu, Aixi Chen

**Affiliations:** Key Laboratory of Optical Field Manipulation of Zhejiang Province, Department of Physics, Zhejiang Sci-Tech University, Hangzhou 310018, China

**Keywords:** quantum battery, hybrid field charging, battery capacity, two-level atom

## Abstract

A quantum battery consisting of an ensemble two-level atom is investigated. The battery is charged simultaneously by a harmonic field and an electrostatic field. The results show that the hybrid charging is superior to the previous case of only harmonic field charging in terms of battery capacity and charging power, regardless of whether the interaction between atoms is considered or not. In addition, the repulsive interaction between atoms will increase the battery capacity and charging power, while the attractive interaction between atoms will reduce the battery capacity and discharge power.

## 1. Introduction

In recent years, based on the progress of quantum information science, researchers have conducted detailed research on various tasks by using quantum information processing [1,2,3,4]. Scientists are more and more curious about the emerging quantum technology that can be applied to new quantum devices. Alicki and Fannes [5] introduced the concept of a quantum battery, which subsequently developed into an important new research field [6,7,8,9,10,11]. The concept of a quantum battery was originally composed of a two-level system, which can temporarily store energy from external transmission [12]. It is found that quantum batteries can also be composed of many different quantum systems, such as the spin system [13], three-level system [14] and magnon-mediated system [15].

Quantum batteries are also generally considered as a series of two-level systems [13,16,17,18], such as the spin chain model, which can effectively increase battery power through spin–spin interaction [13]. In this kind of quantum battery, when the battery is fully charged, if the charging field is not closed, the coherent oscillation of the system will lead to spontaneous discharge, so the battery will bounce back and forth between the charging state and the grounding state. Santos et al. [14] exploited a stimulated Raman adiabatic passage [19] to ensure the stable charged state of a three-level quantum battery which allows one to avoid the spontaneous discharging regime. Like the lithium-ion battery [20,21], capacity and power are two important performance indicators of quantum battery. In addition, a small size is also a key indicator of batteries [22]. Quantum batteries are expected to achieve large capacity and charging power, as well as small battery size. Quach and Munro [23] introduced an open quantum battery protocol, which uses dark states to achieve super extended capacity and power density, and couples non-interacting spins to the energy storage. In quantum information theory, it is well-known that correlation and entanglement can lead to limitations in energy-extraction tasks [24,25,26,27,28]. Therefore, people naturally face a frustrating situation, that is, quantum correlation has both positive and negative effects on the energy storage process. On the one hand, quantum correlation can accelerate the charging time of the quantum battery, and on the other hand, it can impose serious restrictions on the work that can be actually extracted from it.

Recent work mainly focuses on the impact of quantum correlation on the quantum heat engines [29,30] and the charging of collective quantum batteries [31,32,33,34]. The development of quantum heat engines requires emerging field of quantum thermodynamics. A quantum heat engine which consists of a photon gas inside an optical cavity as the working fluid and quantum coherent atomic clusters as the fuel is proposed and shows that the work output becomes proportional to the square of the number of the atoms [29]. Entangled dimers enable a much broader range of cavity temperature control [30]. The Dicke quantum battery [32,33] utilizes a common cavity as a charger and is coupled with the cavity. It has two kinds of quantum correlation; one is caused by the interaction between the charger and the quantum battery, and the other is due to the intrinsic interaction between quantum batteries. Quantum phase transition is a physical phenomenon with many properties identical to quantum correlation in the interaction system. Studying the characteristics of the quantum battery in different phases may be related to collective charging. An interesting research is to charge *N* two-level atoms by using the semi-classical harmonic field [31]. Compared with the previous static charging field, it shows a larger battery capacity, but the power of the quantum battery is reduced.

In this paper, a hybrid field consisting of a semi-classical harmonic field and a static field is considered to charge a quantum battery composed of *N* two-level atoms. Compared with the previous static or semi-classical harmonic charging field [31], it has substantial improvement in battery capacity and charging power. In addition, the influences of the interaction among atoms on the capacity and charging power of the quantum battery are also investigated.

## 2. Quantum Battery Composed of Non-Interacting Atom

The complete quantum battery system is shown in Figure 1. It consists of an ensemble of independent two-level atoms. When t=0, each atom is in the ground state $|g\rangle$. When t=T, each atom is in the excited state $|e\rangle$ and the quantum battery is fully charged, which can be seen in Figure 1a. We consider the quantum battery to be charged by a hybrid driving field B+Acosωt, which can be seen in Figure 1b. The Hamiltonian of *N* non-interacting atoms can be written as [31]
(1)$$H_{0}=\frac{\Delta }{2}\sum_{i=1}^{N}\sigma_{i}^{z}=\Delta S_{z},$$
where atom operators $S_{z}=\sum_{i}\sigma_{i}^{z}/2$, Δ is the energy level splitting of the two-level atom and we let Δ=1 in the following calculation. The eigenstates of *N* two-level atomic system are Dicke states $|S,m\rangle$. Here, we use a hybrid field composed of semi-classical harmonic field and electrostatic field to charge the battery. The Hamiltonian is
(2)$$\begin{array}{ccc} H_{1} & = & \frac{\left(B+A\cos\omega t\right)}{2}\sum_{i=1}^{N}\sigma_{i}^{x}\\ & = & \left(B+A\cos\omega t\right)S_{x}, \end{array}$$
where *A* and *B* are the driving amplitudes of the harmonic field and electrostatic field, respectively. ω is the modulated frequency. Figure 1b shows that the charging field is turned on at time 0 and turned off at time *T*. During the charging process, the total Hamiltonian for *N* two-level atoms interacting with the hybrid field is $H=H_{0}+H_{1}$. In order to highlight the advantages of hybrid field charging, we compare it with the cases of harmonic field charging and electrostatic field charging, respectively. Initially, each atom is in the ground state $|g\rangle$, and the whole system is in the lowest energy state $|\varphi_{N}\left(0\right)\rangle =$$|N,-N/2\rangle$. The wave function can be obtained by solving the Schrödinger equation
(3)$$i\partial |\varphi_{N}\left(t\right)\rangle /\partial t=H|\varphi_{N}\left(t\right)\rangle .$$

At time *t*, the capacity of the quantum battery can be written as
(4)$$\begin{array}{ccc} E_{N}\left(t\right)= & & \langle \varphi_{N}\left(t\right)|H_{0}|\varphi_{N}\left(t\right)\rangle \\ & & -\langle \varphi_{N}\left(0\right)|H_{0}|\varphi_{N}\left(0\right)\rangle . \end{array}$$

The quantum battery can transfer energy from a charging field and store it in atoms. When each atom is in the excited state $|e\rangle$, the system is in a fully charged state, as shown in Figure 1a.

For simplicity, we limit N=1. We regard the Hamiltonian of the electrostatic field as a perturbation term. Therefore, the Hamiltonian of the system is divided into two parts:(5)$$H=H_{a}+H^{{}^{\prime }}=\left(\Delta S_{z}+A\cos\omega tS_{x}\right)+BS_{x}.$$

We use two unitary transformations to approximate $H_{a}$ as a time-independent Hamiltonian [31] with $H_{a}^{i}=U_{i}H_{a}U_{i}^{†}+iU_{i}\frac{d}{dt}U_{i}^{†}$. The two unitary operators are $U_{1}=\exp\left[i\frac{A}{\omega }\xi \sin\omega tS_{x}\right]$ and $U_{2}=\exp\left[i\omega tS_{z}\right]$. $H_{a}^{2}=\Delta_{1}S_{z}+4A_{1}S_{x}$, where $\Delta_{1}=\Delta J_{0}\left(\frac{A}{\omega }\xi \right)-\omega$ and $A_{1}=\frac{A}{2}\left(1-\xi \right)$. The eigenvalues of $H_{a}^{2}$ are $\epsilon_{\pm }^{0}=\pm \frac{\Omega }{2}$ with $\Omega =\sqrt{\Delta_{1}^{2}+4A_{1}^{2}}$. The corresponding eigenstates are
(6)$$\begin{array}{cc} {|\epsilon_{+}\rangle }^{0} & =\sin\theta |g\rangle +\cos\theta |e\rangle \\ {|\epsilon_{-}\rangle }^{0} & =\sin\theta |e\rangle -\cos\theta |g\rangle , \end{array}$$
with $\tan\theta =2A_{1}/\Delta_{1}$. Next, one can solve the eigenvalues of *H* by using perturbation theory as
(7)$$\epsilon_{\pm }=\epsilon_{\pm }^{0}+\epsilon_{\pm }^{1}+\epsilon_{\pm }^{2},$$
where $\epsilon_{\pm }^{1}=$${}^{0}\langle \epsilon_{\pm }|H^{{}^{\prime }}{|\epsilon_{\pm }\rangle }^{0}=\pm B\sin2\theta$, $\epsilon_{\pm }^{2}=\frac{{|}^{0}\langle \epsilon_{\mp }|H^{{}^{\prime }}{|\epsilon_{\pm }\rangle }^{0}{|}^{2}}{\epsilon_{\pm }^{0}-\epsilon_{\mp }^{0}}=\pm \frac{B^{2}\cos^{2}2\theta }{\Omega }$. The corresponding eigenstates are
(8)$$|\epsilon_{\pm }\rangle ={|\epsilon_{\pm }\rangle }^{0}+{|\epsilon_{\pm }\rangle }^{1},$$
with ${|\epsilon_{\pm }\rangle }^{1}=\frac{{}^{0}\langle \epsilon_{\mp }|H^{{}^{\prime }}{|\epsilon_{\pm }\rangle }^{0}}{\epsilon_{\pm }^{0}-\epsilon_{\mp }^{0}}{|\epsilon_{\mp }\rangle }^{0}$. It can be seen from Figure 1a that the initial state $|\varphi (0)\rangle$ of the system is $|g\rangle$. Therefore, we can express the initial state by the linear superposition of the system eigenstates as $|\varphi (0)\rangle =c_{1}|\epsilon_{+}\rangle +c_{2}|\epsilon_{-}\rangle$. At the end of the charging protocol, the final state of the one-atom battery is given explicitly by the eigenstates and eigenvalues
(9)$$|\varphi (t)\rangle =c_{1}e^{-i\epsilon_{+}t}|\epsilon_{+}\rangle +c_{2}e^{-i\epsilon_{-}t}|\epsilon_{-}\rangle .$$

We can obtain the corresponding stored energy of the single-atom battery by substituting Equation (Equation 9) with Equation (Equation 4).

In addition to the battery capacity, it is also important to evaluate the charging power of the quantum battery [35]. The charging power of the present quantum battery is
(10)$$P\left(t\right)=E_{1}\left(t\right)/\Delta t.$$

In the following, we will compare the capacity and power of the quantum battery charged by the hybrid field with the case charged by harmonic and electrostatic fields, respectively. Figure 2 shows that the stored energy $E_{1}\left(t\right)/\Delta$ has maximum value at several peaks. Obviously, $E_{1}\left(t\right)/\Delta$ fluctuates between 0 and 1, which is the discharge caused by the spontaneous transition of the atom from the excited state to the ground state. The maximum of the battery capacity charged by the hybrid field is greater than that charged by the harmonic field and the static fields. Moreover, it is obvious that it takes a short time to fully charge the battery in the hybrid field.

Figure 3 shows the more critical results. It can be seen that the power of the hybrid field charging is much higher than the other two. Therefore, the quantum battery based on hybrid field charging can obtain higher charging power without sacrificing the battery capacity under appropriate parameters.

In order to study the situation when harmonic field and static field are mixed in different proportions, in Figure 4, we show the change of the battery capacity with A/B and charging time *t*. It can be seen from Figure 4 that for a fixed ratio A/B, the battery capacity changes approximately periodically with the charging time. When *A* is larger, the battery capacity is larger, and it can reach the maximum capacity in a short time, that is, it has a larger charging power. However, when A/B>11, the maximum capacity and the charging power become smaller. When 1<A/B<5, the battery has larger capacity and power, the better quantum battery can be obtained.

## 3. Quantum Battery Composed of Interacting Atoms

For *N* identical two-level atoms, long-range forces among all atoms can be mediated by the electric field. Due to the dipole–dipole interaction, the Hamiltonian of *N* interacting atoms can be written as [31]
(11)$$H_{0}^{I}=\frac{\Delta }{2}\sum_{i=1}^{N}\sigma_{i}^{z}+\frac{g}{2N}\sum_{i\neq j}^{N}\left(\sigma_{i}^{x}\sigma_{j}^{x}+\sigma_{i}^{y}\sigma_{j}^{y}\right),$$
where *g* is the atom–atom coupling strength, including the repulsive (g>0) and attractive (g<0) interactions. The total work output from the battery is defined as
(12)$$\begin{array}{ccc} E_{N}\left(t\right) & = & E_{final}-E_{initial}\\ & = & Tr\left[H_{0}^{I}\rho \left(t\right)\right]-Tr\left[H_{0}^{I}\rho \left(0\right)\right]. \end{array}$$

The initial state ρ(0) evolves into ρ(t) under the action of the local charger U(t) as
(13)$$\rho \left(t\right)=U\left(t\right)\rho \left(0\right)U{\left(t\right)}^{†},$$
where $U\left(t\right)=e^{-iH_{1}t}$. In the following work, we take N=2 as an example to study the effect of interatomic interactions on the quantum battery. The Hamiltonian $H_{0}^{I}$ in this case takes the form
(14)$$H_{0}^{I}=\left[\begin{array}{cccc} \Delta & 0 & 0 & 0\\ 0 & 0 & \frac{g}{2} & 0\\ 0 & \frac{g}{2} & 0 & 0\\ 0 & 0 & 0 & -\Delta \end{array}\right].$$

Because the eigenstates about the 4th order matrix are more difficult to find, we change to another equivalent method to solve the battery capacity. We output a time evolution operator through the external field, which acts on the initial density matrix of the system to obtain the density matrix at time *t*. Then we can calculate the battery capacity by finding the trace [36]. After the evolution according to $H_{1}$, the evolved state at time *t* is
(15)$$\rho \left(t\right)\;\;=\;\;\;\;\left[\;\;\begin{array}{cccc} \frac{\beta^{2}}{4}\;\; & \;\;\frac{-i\beta \;\sin\;2A_{1}t}{4}\;\; & \;\;\frac{-i\beta \;\sin\;2A_{1}t}{4}\;\; & \;\;-\frac{\alpha \beta }{4}\\ \frac{-\beta \;\sin\;2A_{1}t}{4i}\;\; & \;\;\frac{\sin^{2}\;2A_{1}t}{4}\;\; & \;\;\frac{\sin^{2}\;2A_{1}t}{4}\;\; & \;\;\frac{\alpha \;\sin\;2A_{1}t}{4i}\\ \frac{-\beta \;\sin\;2A_{1}t}{4i}\;\; & \;\;\frac{\sin^{2}\;2A_{1}t}{4}\;\; & \;\;\frac{\sin^{2}\;2A_{1}t}{4}\;\; & \;\;\frac{\alpha \;\sin\;2A_{1}t}{4i}\\ -\frac{\alpha \beta }{4}\;\; & \;\;\frac{i\alpha \;\sin\;2A_{1}t}{4}\;\; & \;\;\frac{i\alpha \;\sin\;2A_{1}t}{4}\;\; & \;\;\frac{\alpha^{2}}{4} \end{array}\;\;\right],$$
where $\alpha =1+\cos2A_{1}t$ and $\beta =1-\cos2A_{1}t$. The final stored energy reads as
(16)$$E_{2}\left(t\right)=-\cos2A_{1}t+\frac{1}{4}g\sin^{2}2A_{1}t.$$

The relationship between stored energy $E_{2}\left(t\right)/\Delta$ and charging time *t* is shown in Figure 5. g=0 indicates that there is no interaction between atoms. It is clear that the repulsive interaction among the atoms (g>0) can improve the battery capacity, which is better than non-interacting atoms in terms of charging. For attractive interaction scenarios (g<0), it has a negative influence on the energy storage of quantum battery. The stronger the repulsive interaction among the atoms, the more energy of the quantum battery can be stored. This is because the repulsive force among the atoms will increase the energy level spacing of atoms, so that quantum batteries can store more energy.

Figure 6 depicts the charging power $P_{2}\left(t\right)/\Delta$ versus charging time *t* for different *g* with A=B=1. The repulsive interaction among the atoms significantly enhances the power of the quantum battery. The attractive interaction among the atoms will reduce the charging power. Figure 4 and Figure 5 show that the interaction among the atoms can significantly affect the performance of the quantum battery, especially the charging power. In the actual quantum battery, there are interactions among the atoms which should not be ignored. In the actual preparation of the quantum battery, the interaction among the atoms should be modulated into repulsive interaction as much as possible, which can improve the performance of the quantum battery.

Different charging fields will also affect the atomic quantum battery with interaction. Figure 7 shows the stored energy $E_{2}/\Delta$ versus the charging time *t* for different charging fields with g=1 and A=B=1. Although the quantum battery has the same maximum battery capacity under the three charging fields, the quantum battery can reach the maximum capacity quickly under the hybrid field charging. Figure 8 clearly shows that the charging power of the hybrid field charging is almost twice that of the other two cases, which is similar to the case in Figure 3 when the interaction among the atoms is not considered.

We also investigate the stored energy $E_{2}/\Delta$ versus A/B and charging time *t* in Figure 9. Similar to the case in Figure 4 without considering the interatomic interaction, when A/B<15, larger *A* can obtain larger battery capacity and charging power. Different from ignoring the interaction among the atoms, when *A* is small, for example, when A/B is about 1, although the charging power is smaller, the maximum capacity of the quantum battery can be stable for a long time.

The above results show that whether the interaction among the atoms is considered or not, the hybrid field charging is better than the case of harmonic field and static field charging alone, especially in the charging power.

## 4. Conclusions

In this paper, a hybrid field consisting of a semi-classical harmonic field and a static field is proposed as the energy charger of a two-level atom quantum battery. Without considering the interaction among the atoms, hybrid field charging can make the quantum battery have larger capacity and charging power. When considering the interaction among the atoms, we find that the repulsive interaction among the atoms can increase the capacity and charging power of the quantum battery. On the contrary, the attractive interaction among the atoms will reduce the battery capacity and charging power. In addition, the influence of hybrid field charging on the battery capacity is not significant when considering the interaction among the atoms, but the influence on charging power is still significant. We think that our present scheme of hybrid field charging can increase the battery capacity and accelerate the charging speed under the same two-level system.

Recently, the experimental research of quantum battery has also made progress. The advantages and limitations of different profiles for classical drives used to charge these miniaturized batteries have been investigated [37]. The first experiment of charging and self discharging process of quantum battery was realized [38]. The charge and discharge of quantum battery also were realized experimentally [39]. Experimental investigation of a quantum battery was carried out by using star-topology NMR spin systems [40]. The concept of “super absorption” was successfully demonstrated in a new study [41]. This is the key idea supporting the quantum battery, marking that the quantum battery may become a reality. In the present scheme, one can control the opening time of harmonic field and static field so that they can be opened and closed at the same time to achieve the combined effect. For example, when the frequency modulated Raman laser beam and the static field drive the two-level system at the same time, the hybrid field charging is realized. Therefore, we think that it is feasible to combine the harmonic field with the static field to charge the quantum battery.

## Figures and Tables

**Figure 1 entropy-24-01821-f001:**
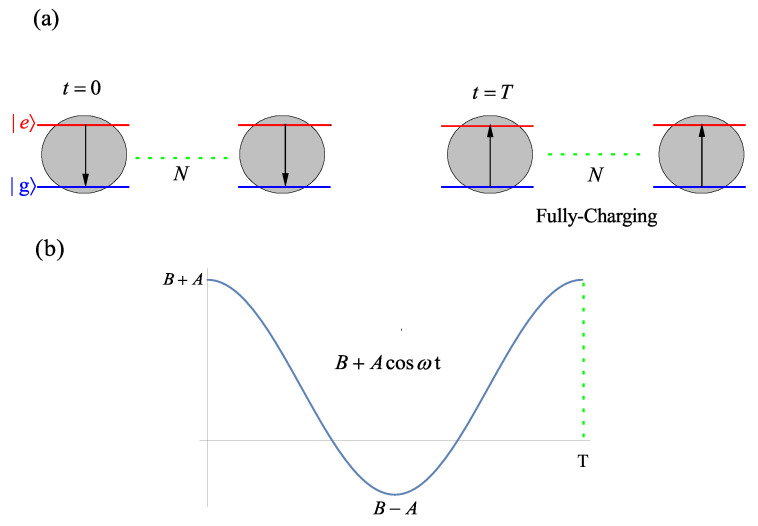
(**a**) Quantum battery system. When t=0, each atom is in the ground state $|g\rangle$. When t=T, each atom is in the excited state $|e\rangle$, and the quantum battery is fully charged. (**b**) The quantum battery is charged by a hybrid driving field B+Acosωt.

**Figure 2 entropy-24-01821-f002:**
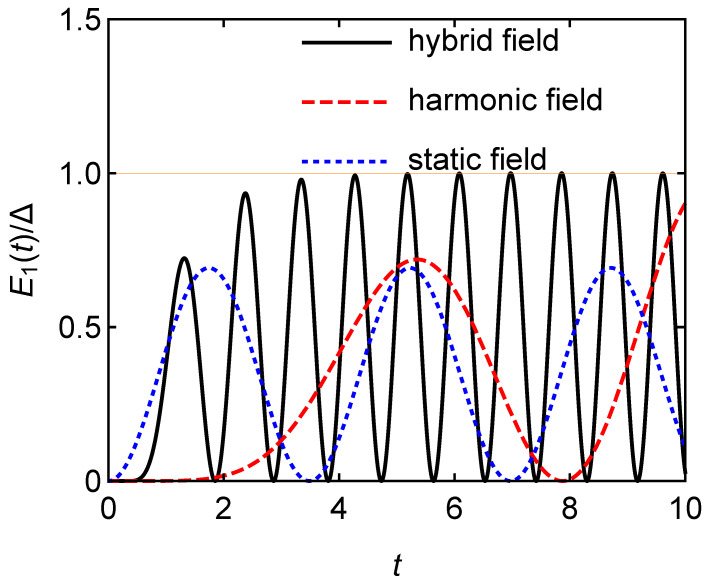
Stored energy $E_{1}\left(t\right)/\Delta$ in the single-atom quantum battery versus charging time *t*.

**Figure 3 entropy-24-01821-f003:**
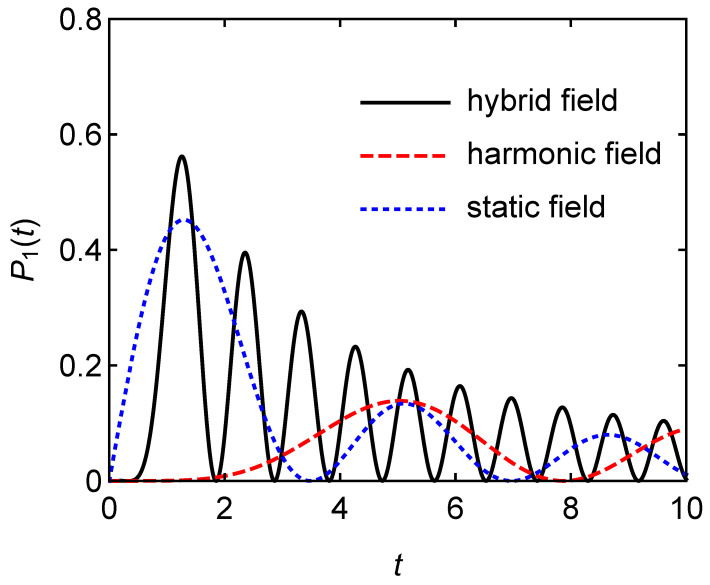
Charging power $P_{1}\left(t\right)$ in the single-atom battery versus charging time *t*.

**Figure 4 entropy-24-01821-f004:**
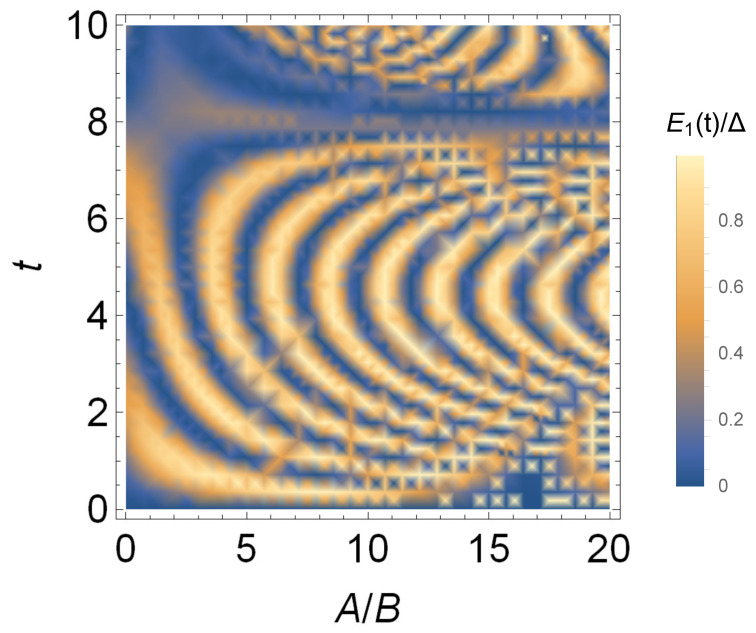
Stored energy $E_{1}\left(t\right)/\Delta$ in the single-atom quantum battery versus A/B and charging time *t*.

**Figure 5 entropy-24-01821-f005:**
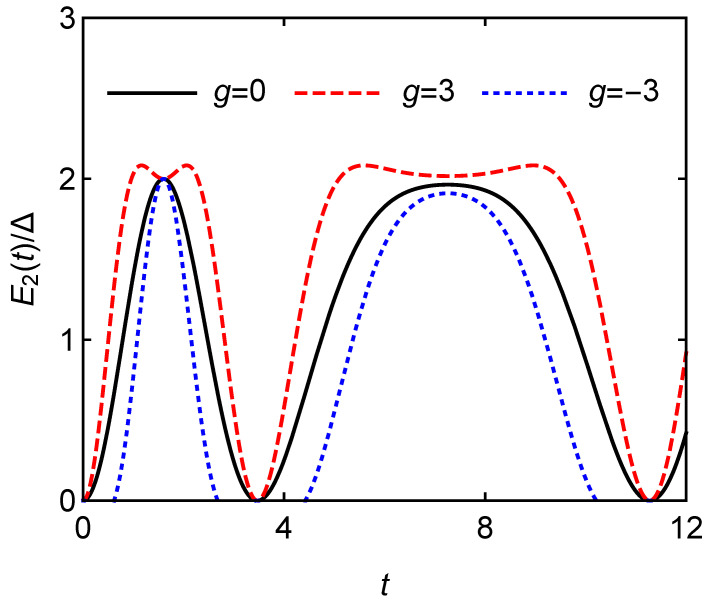
Stored energy $E_{2}\left(t\right)/\Delta$ versus charging time *t* for different *g* with A=B=1.

**Figure 6 entropy-24-01821-f006:**
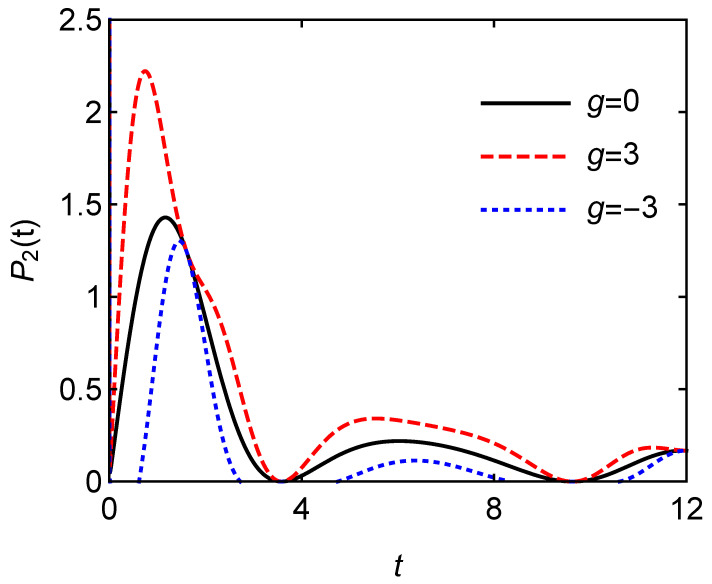
Charging power $P_{2}\left(t\right)$ versus charging time *t* for different *g* with A=B=1.

**Figure 7 entropy-24-01821-f007:**
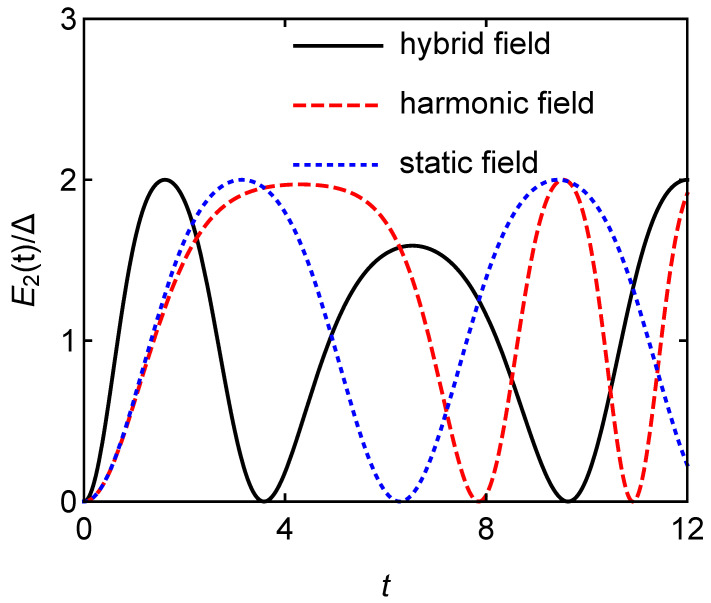
Stored energy $E_{2}/\Delta$ versus the charging time *t* for different charging fields with g=1 and A=B=1.

**Figure 8 entropy-24-01821-f008:**
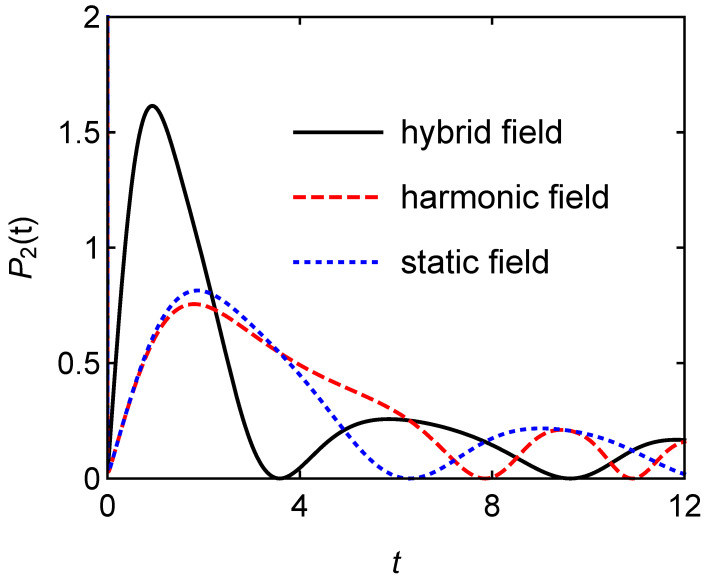
Charging power $P_{2}\left(t\right)$ versus the charging time *t* for different charging fields with g=1 and A=B=1.

**Figure 9 entropy-24-01821-f009:**
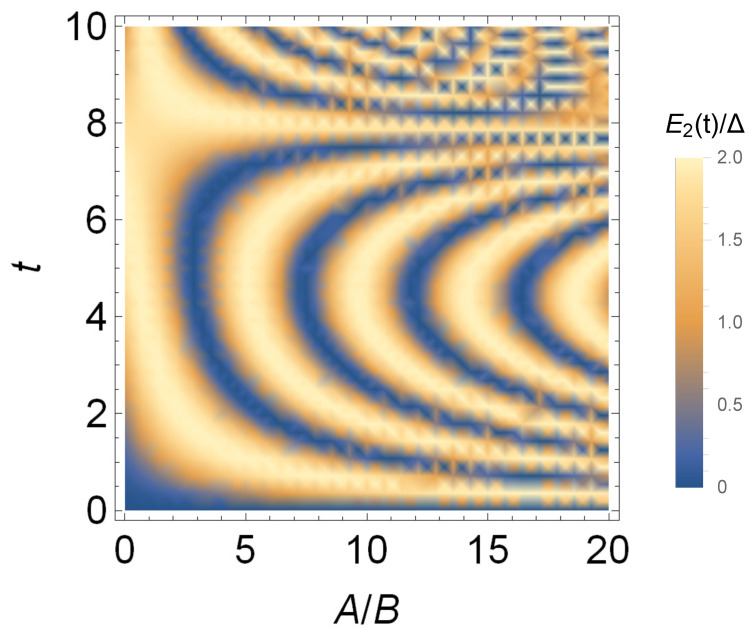
Stored energy $E_{2}/\Delta$ versus A/B and charging time *t* with g=1.

## Data Availability

Not applicable.

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
