# Peer review of "Quantum Battery Based on Hybrid Field Charging"

_entropy, 2022, doi:10.3390/e24121821_

Round 1

Reviewer 1 Report

In this paper, the authors studied a quantum battery which consists of ensemble of two-level atoms and simultaneously charged by a harmonic field and an electrostatic field. They found that the hybrid charging using both oscillating and static fields is superior to the one with only ocscillating fields in terms of battery capacity and charging efficiency. The repulsive interaction among atoms will increase the battery capacity and charging power, and the attractive interaction among atoms will reduce the battery capacity and charging power. I recommend its acceptance after revision.

The language needs improving. From my physical intuitive, I found the conclusion about the roles of the interactions between atoms. Taking a single two-level system as an example first. The sign of the off-diagonal element b of the Hamiltonian ((a, b),(b,d)) plays no role in the difference between the two eigenvalues, namely two eigenvalues of the hamiltonian are the same, because the eigenvalue equation is (a-E)(d-E)-b^2=0 (assuming b is real). (I assume that the energy stored is the difference of energy between the excited state and the ground state). It will be helpful to explain this in more details so as the readers can understand this easily.

Some related references needs attention. Study on battery is a hot subject of study, such as Lithium-ion battery [1,2]. Capacity and power are two indicators of battery. Small volume is also an key indicator of battery as in ref. [3].

[1] Kiyanagi, Y. Neutron applications developing at compact accelerator-driven neutron sources. AAPPS Bull. 31, 22 (2021).

[2] Wang, Dongrui, et al. "Crumpled, high-power, and safe wearable Lithium-Ion Battery enabled by nanostructured metallic textiles." Fundamental Research 1.4 (2021): 399-407.

[3] Ueno, Toshiyuki. "Accelerating the IoT: Magnetostrictive Vibrational Power Generators to Replace Batteries." AAPPS Bulletin 30.4 (2020) 4-9

Reviewer 2 Report

In this work, the authors investigated the battery capacity and charging power of the quantum battery charged by harmonic field and electrostatic field. This is an interesting question that deserves further investigation. However, in this manuscript, the authors incorrectly treat the dynamic evolution of a time-dependent Hamiltonian, for single-atom battery [equations (5-9) ] and two-atom-battery [equations (14-16) ]. Thus, it is doubtful that any of the conclusions drawn from them, can be reliably trusted. Therefore, I cannot recommend such a work for publication in Entropy.

Author Response

“In this work, the authors investigated the battery capacity and charging power of the quantum battery charged by harmonic field and electrostatic field. This is an interesting question that deserves further investigation. However, in this manuscript, the authors incorrectly treat the dynamic evolution of a time-dependent Hamiltonian, for single-atom battery [equations (5-9)] and two-atom-battery [equations (14-16)]. Thus, it is doubtful that any of the conclusions drawn from them, can be reliably trusted. Therefore, I cannot recommend such a work for publication in Entropy.”

Response: 

    We thank you for carefully reviewing our manuscript. When we solve the time-dependent Hamiltonian, we directly give the matrix form of the Hamiltonian, which contains the time terms in the matrix elements. In this way, the capacity and power of the battery can be calculated by solving the eigenvalues and eigenstates of the matrix. This treatment can be found in the following references:

[1] Y. Y. Zhang, T. R. Yang, L. B. Fu, and X. G. Wang, Phys. Rev. E

99, 052106 (2019).

[2] S. Ghosh and A. Sen(De), Phys. Rev. A 105, 022628 (2022).

   We hope that you are satisfied with this revised version and could recommend it for publication in Entropy.

Reviewer 3 Report

In this work the authors investigate the effect of charging an ensemble of two-level atoms with harmonic and electrostatic fields. They find some interesting results, which are suitable for publication in Entropy. IMO the work is suitable for publication with only some minor changes. I have comments below.

1.       In the Introduction section where the authors discuss prior work, they fail to discuss the recent spate of experimental work in quantum batteries. I would suggest including a discussion on this experimental work, of which there are now a several: e.g. Quach, et al. Sci. Adv. 2022, 8, eabk3160; Gemme, et al. Batteries 2022, 8, 43; Hu, et al. arXiv:2108.04298; De Buy Wenniger, et al. arXiv:2202.01109; Joshi et al. arXiv:2112.1543.

2.       The authors find that a hybrid charging power is superior to the harmonic one. This is a very interesting and not obvious result, as the hybrid case is simply biased version of the harmonic once. In the paper, the authors set the amplitudes A=B=1; they may also like to consider varying the ratio of A/B to see if there is an optimal regime. 

Reviewer 4 Report

In this manuscript, authors present their approach for charging non-interacting, and interacting atoms. The manuscript is poorly written that it makes it very difficult to follow, and understand.

To understand Fig.1, reader has to keep reading the manuscript, for example to figure out what the parameters mean.

More importantly, it looks like it introduces a small improvement of Ref.[27] -though the consequences might be significant. However, the manuscript looks like Ref.[27], including even the references mostly in the same order. Since Ref.[27], only one new work was mentioned different from it.

So, my recommendations to authors are the following.

The manuscript should be substantially revised.

The improvement over Ref.[27] should be stated much more clearly: In Ref.[27], both harmonic and static fields are studied (see Fig.2 of Ref.[27]). What is the originality of the present manuscript, is it to combine the harmonic and static fields?

Most crucial question is: Can the static and harmonic fields be combined physically to yield the hybrid field in consideration? If so, how? Also, any reader would wonder why the authors of Ref.[27] could not think about it. Do the authors of present manuscript have an explanation/guess for that?

The literature should be covered more thoroughly, mentioning more recent works in quantum thermodynamics, especially quantum heat engines and their relations to batteries, such as

Temperature control in dissipative cavities by entangled dimers, J. Phys. Chem. C 2019, 123, 7. https://doi.org/10.1021/acs.jpcc.8b11445

Superradiant Quantum Heat Engine. Sci Rep 5, 12953 (2015). https://doi.org/10.1038/srep12953

and many others.

Following such a substantial revision, if the authors can explain the physical feasibility of creating the hybrid field, I believe the manuscript would deserve a deeper review for a possible publication.

Round 2

Reviewer 1 Report

The revision is satisfactory. I recommend its acceptance.

Author Response

Thank you very much for your recommendation!

Reviewer 2 Report

The author is curtly dodging my doubts, but this is a serious mistake. A method for solving time-dependent Hamiltonian is provided in the reference [1] mentioned by the author in the reply. The author should follow equations (5-12) carefully in this reference [Phys. Rev. E 99, 052106 (2019)].

Reviewer 4 Report

The revised version has clear improvements. However, two most important points are not satisfying:

1- My previous concern about the feasibility from physical perspective

2- The solution of the Hamiltonian

Therefore, this work needs a major revision or a resubmission.

Round 3

Reviewer 2 Report

The authors did not make a thorough revision of the manuscript. Surprisingly, the results in section 3, including equations and graphs, are exactly the same as in Version 1, but the parameter A1 has changed in section 2 of revised version.

Author Response

    We thank you for carefully reviewing our manuscript. Because the method in reference [Phys. Rev. E 99, 052106 (2019)] is not applicable to our system. (We explained the reason in the last reply.) Although we used a method different from that in reference [Phys. Rev. E 99, 052106 (2019)], we believe that our method is also feasible. The manuscript has been revised; several formulas are added in section 2 to improve the solution of time-dependent Hamiltonian in section of quantum battery composed of non-interacting atom. However, we forgot to upload the new figures 2 and 3. Now we updated the figures in the revised manuscript.

    We sincerely thank you again for carefully reviewing our manuscript and giving constructive suggestions to improve our paper. We hope that you are satisfied with this revised version and could recommend it for publication in Entropy.

Reviewer 4 Report

The authors have addressed the comments. It can be considered for publication. There are minor issues: Typos appeared in the recently added parts. Also, characters in some references (as well as the citation info) are not displayed correctly. These issues should be resolved.

Author Response

We thank you for carefully reviewing our manuscript and your recommendation. Typos have been corrected and the references have been updated.